# Quantitative analysis of Weibel-Palade bodies

**Alice Liu** [1,2], **Christopher J. Ng** [2] *

**1** Department of Bioengineering, Washington University, St. Louis, MO, United States of America,
**2** Department of Pediatrics, University of Colorado–Anschutz Medical Campus, Aurora, CO, United States of America

* Christopher.ng@cuanschutz.edu

**Data Availability Statement:** All relevant data are within the paper and its Supporting Information files.

**Funding:** This work was supported by grants/support from the National Hemophilia Foundation (CJN), the Health Resources & Services

## Abstract

Weibel Palade bodies (WPBs) are vesicles found in endothelial cells which carry the multimeric protein von Willebrand factor (VWF). As cellular confluency has been shown to influence the number of WPBs in endothelial cells, we propose to test two methods of attaining endothelial cell confluence to inform on the relevancy of cellular culture methods when analyzing endothelial WPBs. We test these cellular culture methods in two endothelial cell types, human umbilical vein endothelial cells (HUVECs) and endothelial colony forming cells (ECFCs). One method maintains a constant incubation time of 96 hrs. while varying the seeding density. The second method maintains a constant seeding density of 30,000 cells/cm$^2$ while varying incubation time. In comparing these two methods, we evaluate the nuclei count, total WPB count, and WPB/nuclei count for each. Our results show that there is a trend of increasing nuclei count, total WPB count, and WPB/nuclei count as incubation time and seeding density increases. However, there is no difference in WPB/nuclei quantification whether confluency is reached via a constant seeding density or a constant incubation time. In addition, we show that confluency plays a major role in WPB/nuclei generation as we demonstrate higher WPB/nuclei counts in confluent cultures compared to sub-confluent cultures.

## Introduction

Von Willebrand factor (VWF) is a plasma protein that plays an essential role in hemostasis by mediating platelet adhesion and aggregation [1]. Von Willebrand Disease (VWD) is a bleeding disorder caused by quantitative and qualitative defects in VWF [2]. VWF is synthesized in endothelial cells, where it is assembled into dimers in the endoplasmic reticulum and further assembled into multimers upon arriving in the trans-Golgi network (TGN) where it finishes the maturation process [1, 2]. Highly multimerized forms of VWF are stored in Weibel Palade bodies (WPBs), which are elongated secretory organelles found in endothelial cells [3, 4]. Though VWF is necessary in order for WPBs to form, WPBs also store a variety of other components such as P-selectin, interleukin-8, and endothelin [5]. Morphologically, WPBs are long, rod-shaped, cylindrical bodies with a diameter of 0.1–0.3 μm and a length of 1–5 μm [3].

It has been previously demonstrated that endothelial cell confluence regulates Weibel-Palade body formation [2]. Howell, et al. showed an increasing confluency of an endothelial

Administration – Maternal & Child Health (5 H30 MC00008-20-00 to CJN), the National Institutes of Health (P01 HL144457 to CJN). There was no additional external funding received for this study. Sponsors and funders had no role in study design, data collection and analysis, decision to publish, or preparation of manuscript.

**Competing interests:** The authors have declared that no competing interests exist.

monolayer is associated with higher quantities of WPBs. However, it has not been established whether increasing seeding density or incubation time (both of which can achieve confluency) alter WPB quantification [6]. Our study aims to elucidate the effects of how seeding density or incubation time affect confluency and WPB quantification. With this knowledge, investigators will have a better understanding of the experimental conditions that affect WPB quantification which may allow them to optimize time as well as reagents. Our hypothesis is that WPB quantity is not dependent on the method to achieve confluency. This study adds experimental knowledge to plan experiments most efficiently based on quantification of time to confluency. To confirm the assertion that WPB quantification is not dependent on seeding density or incubation time, we measure WPB quantity after using two methods of achieving confluency.

As VWF is produced primarily by endothelial cells [7], we used both human umbilical vein endothelial cells (HUVECs) as well as endothelial colony forming cells (ECFCs) to test our hypothesis. HUVECs are cells derived from the endothelium of the umbilical cord and are a widely used source of primary endothelial cells for *in vitro* studies. In addition to their availability, ease of culture, and established characterization, HUVECs have also been shown to contain VWF, delineating endothelial cells from vascular smooth muscle cells and fibroblasts [8]. Though HUVECs are extensively used, they do have their limitations. As these cells are derived from a tissue source that is not found in non-neonatal individuals, their relevance to endothelial physiology is at times questioned. To better address endothelial cells from adult individuals, we also use ECFCs derived from peripheral whole blood [9]. ECFCs have been used in various aspects of endothelial research [9], and are phenotypically similar to vascular endothelial cells, exhibiting a cobblestone morphology, and express endothelial cell markers such as VE-cadherin, endoglin, and PECAM-1 [10]. Previous research has demonstrated that these cells display abnormalities in VWF from patients with VWD, suggesting that they can reproduce abnormalities in VWF biology [11, 12].

## Results/Discussion

### Evaluation of doubling time in HUVECs and ECFCs

The classic morphology of HUVECs/ECFCs was visualized using DAPI staining of the nuclei and WPB were stained with a FITC-labeled anti-VWF antibody (Fig 1A and 1B).

The proliferation of HUVECs has been previously evaluated, measuring proliferation relative to the addition or absence of medium supplements simulating various culture environments [13]. However, no definitive quantitative cell counts were reported [10, 14]. Therefore, in this report, we determined the doubling time of both HUVECs and ECFCs. HUVECs and ECFCs exhibited a mean doubling time of 33 and 26 hrs., respectively. Although they start at different seeding densities, the growth curves of HUVECs and ECFCs were similar and were not statistically different (Fig 1C).

We found that the rates of growth for HUVECs and ECFCs were not statistically different, but the nuclei count of ECFCs were lower than HUVECs at every time point, including the initial seeding. One possible hypothesis for this discrepancy between the two cell lines is different binding affinities for collagen. ECFCs are routinely grown on collagen coated plates [15], suggesting that they may have decreased attachment as compared to HUVECs and this may account for the decreased cell count in initial and subsequent quantification. We note these differences but feel that they do not affect our overall conclusions. While starting from different initial cell counts, both HUVECs and ECFCs attain confluency, produce mature WPB, and demonstrate similar growth rates. Future studies with other HUVEC/ECFC cell lines are likely indicated to re-evaluate this finding.

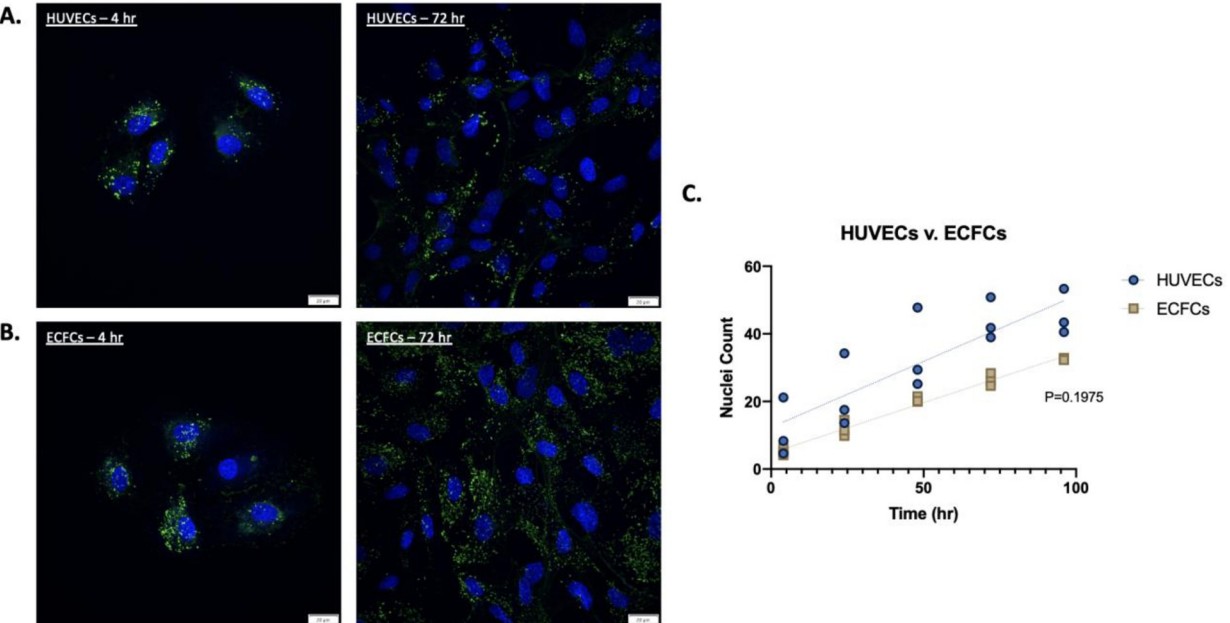

**Fig 1. Morphology and growth rates of HUVECs and ECFCs.** (A) A sample of HUVECs seeded on gelatin coverslips. After 72 hrs. of culture with a seeding density of 30,000 cells/cm$^2$, cells were fixed and stained with ProLong Mountant with DAPI and anti-VWF FITC antibody to visualize nuclei and VWF, respectively. (B) The same was done for ECFCs, showing WPB distribution. (C) Nuclei count of HUVECs and ECFCs were evaluated at varying incubation time points (4, 24, 48, 72, 96 hrs.) with a constant seeding density of 30,000 cells/cm$^2$ (n = 3). The dotted lines represent the growth rate curves as analyzed by a linear regression line of each cell line. The growth rate slopes for HUVECs and ECFCs are 0.39 and 0.30, respectively. The slopes are not significantly different, and P-value shown was determined by linear regression (GraphPad Prism).

### Incubation time analysis: WPB/Nuclei count increases with incubation time for ECFCs but not for HUVECs

We next determined if the WPB/Nuclei count increases with incubation time. For both HUVECs and ECFCs, the average WPB/Nuclei count appeared to significantly increase with increasing incubation time (Fig 2A and 2B). This suggests similarities in WPB formation in ECFCs compared to HUVECs in regard to incubation time.

### Seeding density analysis: Seeding density has no effect on WPB/Nuclei counts

We also examined the effect of seeding density on WPB/Nuclei counts. For HUVECs, the initial seeding density does not affect average WPB/Nuclei counts after our predetermined time point of 96 hrs. While the average WPB/Nuclei count saw a slight increase with increasing seeding density, this change was not statistically significant (Fig 2C and 2D). ECFCs exhibited the same apparent increase in average WPB/Nuclei count with increasing seeding density (Fig 2C), but similar to HUVECs, this change was not statistically significant (Fig 2D). While there was a trend of increase in the number of WPB/Nuclei count with increasing seeding density (similar to that seen by increasing incubation time), we suggest that this is driven by the increased confluency of the cells as previously asserted by Howell et. al. [2].

### Difference in WPB/Nuclei count is statistically significant in sub-confluent vs confluent cell cultures

There have been previous studies exploring endothelial confluence regulating WPB formation as well as the specifics of the formation and function of WPBs. Howell et. Al reported that

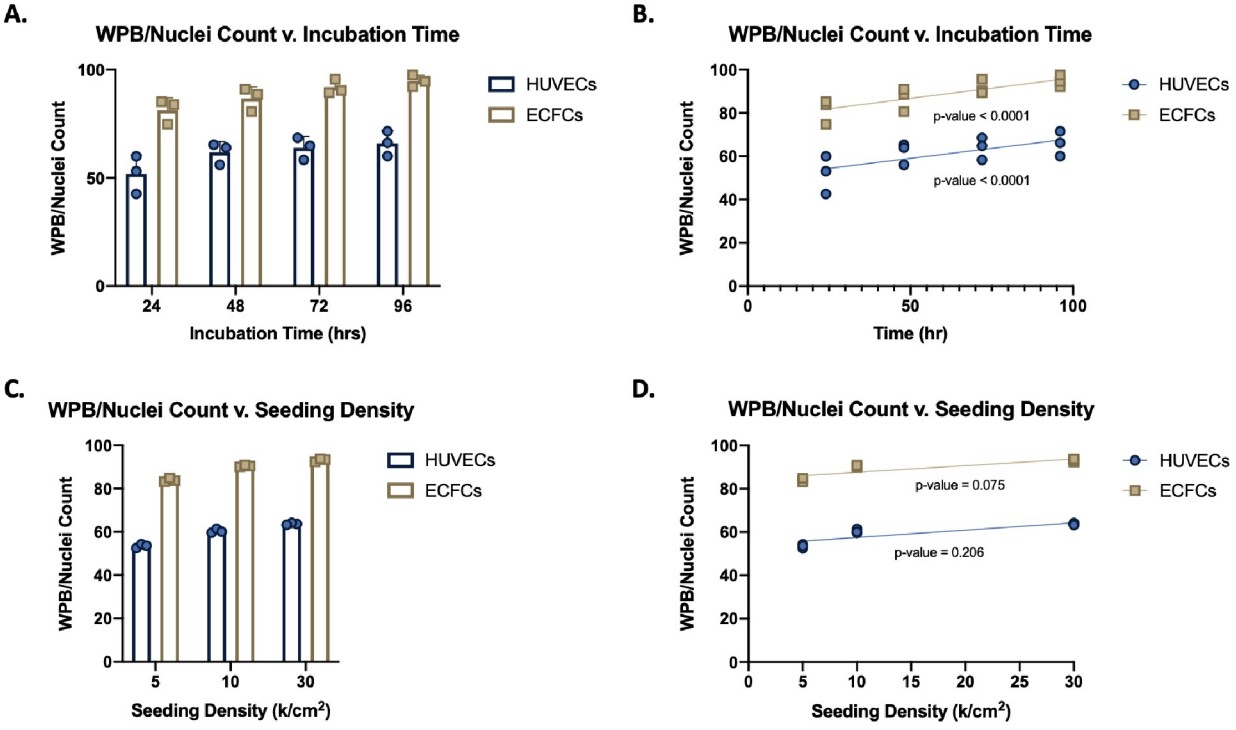

**Fig 2. Effect of incubation time and seeding density on WPB/Nuclei count for HUVECs and ECFCs.** (A) Evaluation of WPB/Nuclei count for both HUVECs and ECFCs in response to increasing incubation time (24, 48, 72, and 96 hrs.) with a constant seeding density of 30,000 cells/cm$^2$ (n = 3). (B) The linear regression analysis of WPB/Nuclei count for a constant seeding density. (C) WPB/Nuclei count is evaluated in response to varying seeding densities (5,000, 10,000, and 30,000 cells/cm$^2$) at a constant incubation time of 96 hrs. (n = 3). (D) The linear regression analysis of WPB/Nuclei count for a constant incubation time.

WPB formation is correlated to the confluency of the endothelial monolayer [2]. Our study expounds on this observation and provides several findings about the effects of confluency on WPB/Nuclei count as well as two different methods of achieving confluency: varying seeding density and varying incubation time.

To assess the effect of confluency on WPB/Nuclei count, cell cultures were divided into two cohorts: sub-confluent and confluent, as defined in the methods section. The difference in WPB/Nuclei count comparing sub-confluent and confluent was found to be statistically significant for both HUVECs and ECFCs. Sub-confluent HUVECs averaged 52.724 WPB/Nuclei whereas confluent cultures averaged 62.95. For ECFCs, sub-confluent cultures averaged 86.893 WPB/Nuclei whereas confluent cultures averaged 93.97 (Fig 3).

Based on our work, we show that confluency is a major driver of WPB/nuclei quantity. Previous studies have reported more mature cytoplasmic WPB as opposed to immature perinuclear WPB to be found in confluent cell cultures [3]. By comparing confluent vs. sub-confluent cultures, we found a similar result of increased WPB/nuclei quantities in confluent cells. One potential reason for this confluent vs. sub-confluent difference could be that there is a minimum threshold of cell confluence before WPB generation reaches its peak, as there are prior reports of WPB formation dependent on endothelial cell-cell contact [2]. The presence of VWF drives the formation of WPBs, but the cellular machinery that controls WPB formation also influences the structural arrangement of VWF [16]. However, recent reports show a contradictory finding that intracellular VWF levels are inversely correlated with the cell nucleus count [17]. Though our findings are conflicting, it is worth noting that Popa et al. evaluates

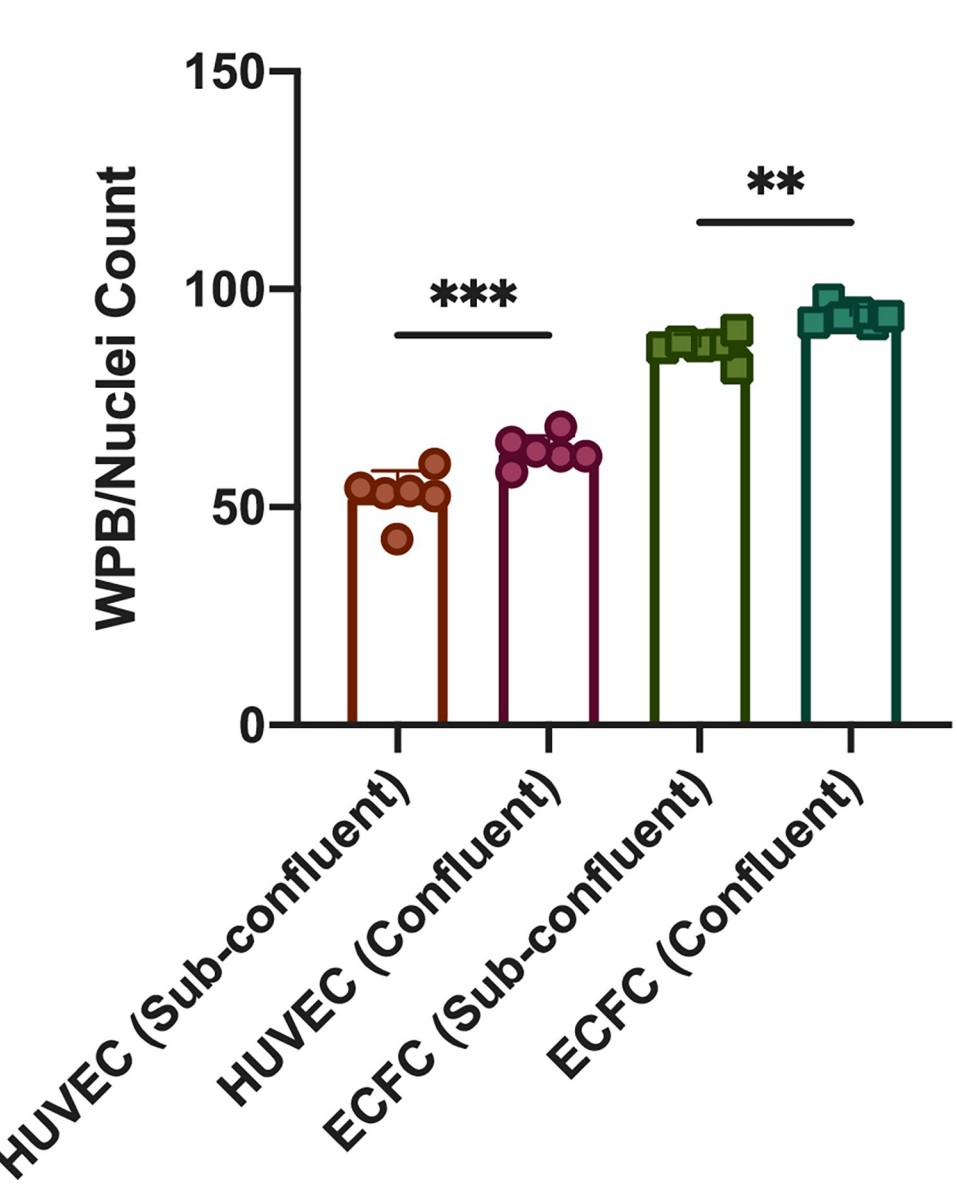

**Fig 3. Assessment of WPB/Nuclei count in sub-confluent versus confluent cultures.** WPB/Nuclei count is assessed in both sub-confluent and confluent cell cultures. Confluence was determined to be 51 cells/field for HUVECs and 75 cells/field for ECFCs as established in methods. A 1-way ANOVA was performed to analyze the WPB/Nuclei count of HUVECs and ECFCs in sub-confluent v. confluent cell cultures, ** $p<0.0021$, *** $p<0.0002$ (n = 3).

total VWF whereas our analysis includes only VWF in WPB. We analyze the total number of WPBs and not the total intracellular protein content of VWF.

Interestingly, we found that ECFCs had a higher WPB/Nuclei count on average as compared to ECFCs. One possible reason to explain this discrepancy is the morphological difference in WPBs of HUVECs compared to those of ECFCs. The size of HUVECs was slightly larger than that of ECFCs, as shown by the finding of 51 cells/field for HUVECs and 75 cells/field for ECFCs at confluency. The method of comparing cell size between HUVECs versus

ECFCs was based off of qualitative of brightfield image analysis (data not provided). Therefore, the larger size of HUVECs may have led to a lower nuclei count, in turn leading to a higher WPB/Nuclei count. In future studies, we should consider staining with VE cadherin to better delineate the cell sizes.

### Comparison of WPB/Nuclei count reaching confluency via incubation time or seeding density

The method of culture to reach confluency was tested to investigate the effect on WPB/Nuclei count. We demonstrate that there is no difference in WPB/Nuclei count for HUVECs or ECFCs using either method (Fig 4). Upon reaching confluency using a constant seeding density (method one) vs constant incubation time (method two), HUVECs averaged 63.85 WPB/Nuclei vs 62.06, respectively. For ECFCs, the average WPB/Nuclei count was 93.21 with method one and 94.73 for method two.

Our current report suggests that it is best to image and examine WPB once confluency is reached for the most biologically accurate assessment of WPBs and that the mechanism to achieve confluency has little effect on the WPB quantification. One limitation of our method is the lack of junctional staining to visualize clear cell morphology. Out method, which utilizes DAPI nuclei staining, should accurately quantify the number of cells but may not completely delineate the cellular junctions and any potential spacing between cells. Another limitation/point of variation in our approach was the definition of confluence. While we defined confluence as 80% of the surface area, we acknowledge that this is a self-defined parameter. In the future, cadherin or other alternative junctional staining should be considered to better delineate cellular borders and to better quantify cellular confluency. We have demonstrated that the method of which we use to reach confluency does not matter as after confluency is achieved, there is no statistical difference in WPB/Nuclei. Therefore, care should be taken to ensure similar levels of confluency prior to comparing various comparative groups. This finding allows for flexibility in experimental planning due to the fact that time can be saved with a higher seeding density, but resources can also potentially be saved with longer periods of incubation.

## Materials and methods

### Reagents

PBS, trypsin, Triton-X, and paraformaldehyde were all from Sigma Aldrich (St. Louis, MO), HUVECs and EBM2-MV Endothelial Cell media were from Lonza (Switzerland), gelatin-coated coverslips were from Neuvitro (Vancouver, WA), ProLong Glass Antifade Mountant was from ThermoFisher Scientific (Waltham, MA), rabbit anti-human VWF polyclonal antibody DAKO (Carpinteria, CA).

### ECFC isolation

Healthy controls are recruited and enrolled in our IRB-approved study at the University of Colorado (COMIRB #15–1072). After enrollment, 50 mL of whole blood is drawn from each individual into sodium heparin tubes (BD Biosciences). The mononuclear cell layer was isolated via centrifugation and plated on collagen-coated tissue culture plates as published. Additional FBS (Hyclone) was added to ECFC EBM2-MV media (Lonza) to attain a final ratio of 18% FBS. Media was changed every two days for up to three weeks or until a colony of endothelial cells was observed. ECFCs exhibited the classical cobblestone appearance of endothelial cells. After the initial outgrowth of ECFCs, the endothelial origin of ECFCs were assayed for cell surface markers (CD31, CD34, CD45, CD105, CD146) by flow cytometry with a Gallios

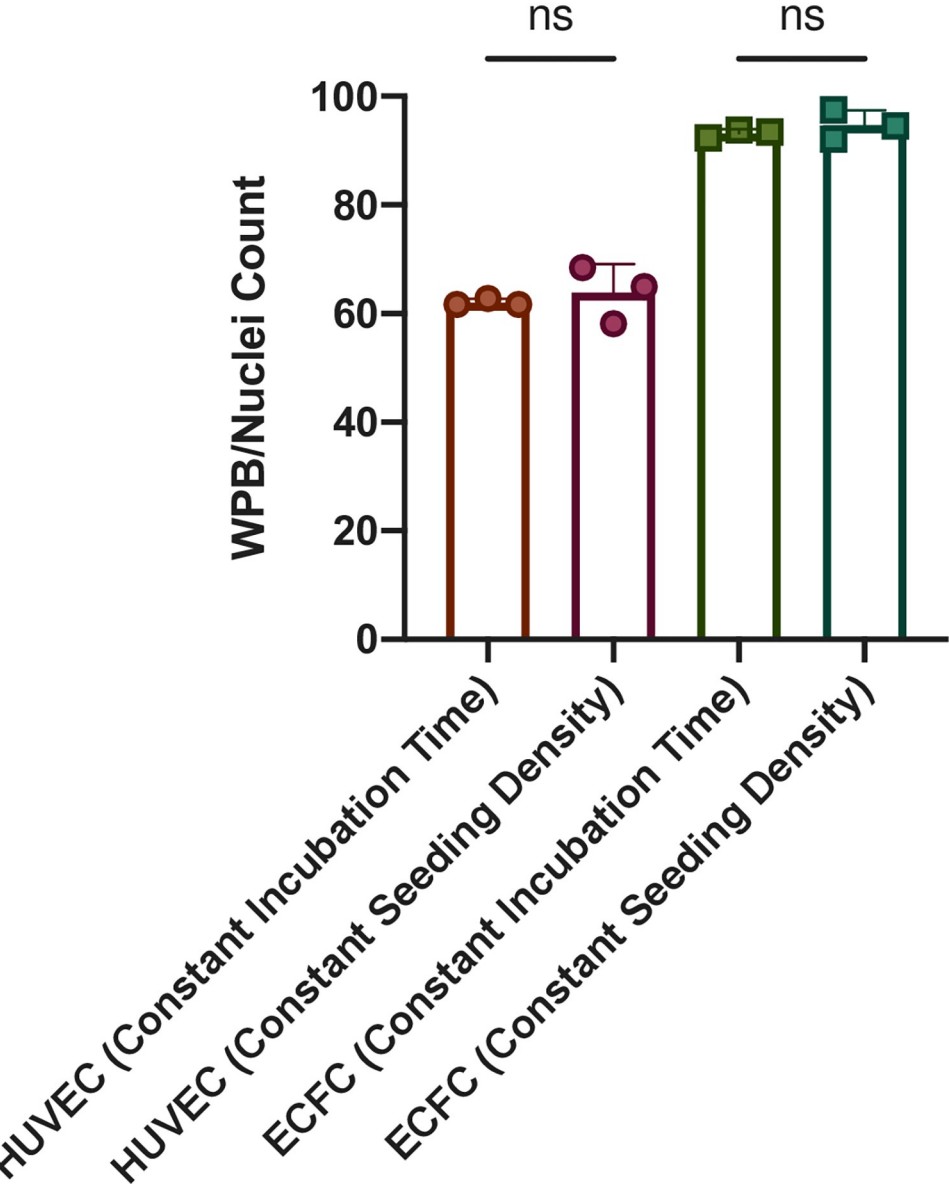

**Fig 4. Comparison WPB/Nuclei count depending on culture method.** WPB/nuclei count for constant incubation time (96 hrs.) and for constant seeding density (30,000 cells/cm$^2$) are shown. A 1-way ANOVA was performed to analyze the WPB/nuclei between constant incubation time and constant seeding density for HUVECs and ECFCs, ns p>0.05 (n = 3).

flow cytometer (Beckman-Coulter). ECFCs were positive for the endothelial markers PECAM (CD31), endoglin (CD105), and negative for leukocyte common antigen (CD45). CD34 exhibited mildly positive expression patterns constituent with other ECFC isolations [12, 18]. Colonies were expanded and aliquoted to be stored in liquid nitrogen. For each experiment, cells were maintained in EBM2-MV with FBS supplemented to 10% (Lonza). All endothelial cells used are passage < 5.

## Cell culture

Human umbilical vein endothelial cells (HUVECs) from Lonza and blood outgrowth endothelial cells (ECFCs) from healthy control individuals were cultured in EBM2-MV with FBS supplemented to 10% HUVECs and ECFCs were plated on gelatin-coated coverslips (Neuvitro), and all experiments used cells passage < 5. For all ECFC-related experiments, two control ECFC cell lines were used.

To compare the methods of achieving confluency, we analyzed two variables. One experiment was done with a constant incubation time of 96 hrs. and varying seeding densities of 5,000, 10,000, and 30,000 cells/cm$^2$ (method one). A second experiment was performed with a constant seeding density of 30,000 cells/cm$^2$ incubated for varying time points of 24, 48, 72, and 96 hrs. (method two). Each experiment was performed a minimum of 3 independent times with a minimum of 5 slides (biological replicates).

After the set amount of incubation, cells were fixed with 4% paraformaldehyde for 15 min followed by washing in PBS (x3). After washing, cells were permeabilized with 0.02% Triton-X for 15 min followed by washing in PBS (x3). Following fixation and permeabilization, cells were incubated for 30 min with Alexa 488-labeled anti-VWF antibody (Bio-Rad) and then washed in PBS (x3). Coverslips were then inverted and mounted onto glass slides using mounting medium with DAPI (ProLong) (Fig 1B).

## WPB imaging

Coverslips prepared as above are inverted onto an Olympus spinning disc confocal microscope (Olympus IXS83-DSU). DAPI and FITC lasers were set to a strength level of 20% with the auto-contrast setting selected. Images are acquired with 23 z-slices with a 0.3 μm step size. A minimum of 5 slides were imaged with 9 locations each, over 3 separate experiments. To minimize observer bias, all 45 locations per experiment were predetermined using a rectangular overview with 9 pre-set imaging locations.

## Doubling time analysis

To determine the doubling time of endothelial cells, HUVECs and ECFCs are seeded at 30,000 cells/cm$^2$, mounted, imaged, and counted at time points of 4, 24, 48, 72, and 96 hrs. With the cell counts at each time point, a doubling time was calculated for each time interval using the following formula [19].

$$Doubling\ Time = \frac{time\ final - time\ initial}{\frac{\log(population\ final) - \log(population\ initial)}{\log(2)}}$$

We defined a confluent cell culture of HUVECs and ECFCs as a single cohesive monolayer of cells covering approximately 80% of the culture surface via a qualitative assessment of immunofluorescent microscopy. Quantitatively, this averages to be 51 cells/field for HUVECs and 75 cells/field for ECFCs on a 60X magnification. We strove to create a quantitative cutoff to minimize variation and bias which can occur if each individual image was analyzed for being "confluent" vs. "non-confluent." Once this definition of confluence was determined, all images were automatically separated using this cutoff into "confluent" or "non-confluent" images. With a seeding density of 30,000 cells/cm$^2$, confluency took 48 and 96 hrs. to achieve confluency in HUVECs and ECFCs respectively.

## Image analysis

Captured images were imported into FIJI [20] to be analyzed for both total number of nuclei as well as WPB via an automated image analysis macro (supplemental macro). TIF images

were split into two separate channels, DAPI and FITC. Each slice image underwent the same processing of auto-contrasting, 8-bit binarization, and auto-thresholding. Using the FIJI plugin "Analyze Particles," nuclei and WPB counts were generated. A custom shell macro was used to calculate and analyze the number of WPB per nuclei (WPB/nuclei) on a per slide basis. As mature WPB have been well characterized as elongated structures varying in size, small round immature WPB were excluded using the circularity feature of Fiji.

## Statistics

Statistical analysis was performed with GraphPad Prism 7.03 (La Jolla, USA). Linear regression analysis and paired T-tests were evaluated, and significance was determined with one-way ANOVA. All data are presented as the mean ± standard error of the mean unless otherwise noted.

## Supporting information

**S1 File.**
(DOCX)

## Author Contributions

**Conceptualization:** Alice Liu, Christopher J. Ng.

**Data curation:** Alice Liu, Christopher J. Ng.

**Formal analysis:** Alice Liu, Christopher J. Ng.

**Funding acquisition:** Alice Liu, Christopher J. Ng.

**Investigation:** Alice Liu, Christopher J. Ng.

**Methodology:** Alice Liu, Christopher J. Ng.

**Project administration:** Alice Liu, Christopher J. Ng.

**Resources:** Alice Liu, Christopher J. Ng.

**Software:** Alice Liu, Christopher J. Ng.

**Supervision:** Alice Liu, Christopher J. Ng.

**Validation:** Alice Liu, Christopher J. Ng.

**Visualization:** Alice Liu, Christopher J. Ng.

**Writing – original draft:** Alice Liu, Christopher J. Ng.

**Writing – review & editing:** Alice Liu, Christopher J. Ng.

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
