## [Decision Letter · Decision Letter 0]

8 Mar 2022

PONE-D-22-04392Quantitative Analysis of Weibel-Palade BodiesPLOS ONE

Dear Dr. Liu,

Thank you for submitting your manuscript to PLOS ONE. After careful consideration, we feel that it has merit but does not fully meet PLOS ONE’s publication criteria as it currently stands. Therefore, we invite you to submit a revised version of the manuscript that addresses all the points raised during the review process.

 Two experts have evaluated the manuscript and agreed that further amanements are needed including the following points:- a clear hypothesis and justification of the study is needed, especially, that WBP density in cutlured endothelial cells is not t new idea and topic. What does the study add to the field in terms of new knwoledge?- Please clearly indicate the number of biological paralllels as well as the number of independent experiments.- add individual data points to figures (in addition to the mean)- junctional staining should be also performed to clearly show cell morphology

We look forward to receiving your revised manuscript.

Kind regards,

Mária A. Deli, M.D., Ph.D.

Academic Editor

PLOS ONE

Journal Requirements:

(This work was supported by grants/support from the National Hemophilia Foundation (CJN), the Health Resources & Services Administration – Maternal & Child Health (5 H30 MC00008-20-00 to CJN), the National Institutes of Health (P01 HL144457 to CJN)

https://www.hemophilia.org

https://mchb.hrsa.gov

https://www.nhlbi.nih.gov

Sponsors and funders had no role in study design, data collection and analysis, decision to publish, or preparation of manuscript.)

Reviewers' comments:

Reviewer's Responses to Questions

**Comments to the Author**

1. Is the manuscript technically sound, and do the data support the conclusions?

Reviewer #1: Yes

Reviewer #2: Partly

2. Has the statistical analysis been performed appropriately and rigorously? 

Reviewer #1: Yes

Reviewer #2: Yes

3. Have the authors made all data underlying the findings in their manuscript fully available?

Reviewer #1: Yes

Reviewer #2: No

4. Is the manuscript presented in an intelligible fashion and written in standard English?

Reviewer #1: Yes

Reviewer #2: Yes

5. Review Comments to the Author

Reviewer #1: Lui et al have investigated how endothelial cells reaching confluence determines their WPB load. Using HUVECs and ECFCs they find that endothelial cells, when given the opportunity to form confluent monolayers will reach some sort of a plateau in terms of WPB numbers per cell. This confirms data from Howell et al from >15 years ago. Furthermore, they show that the route taken to in vitro confluence, be it higher initial seeding density or prolonged incubation time, has hardly any influence on the WPB plateau. This paper lacks a clear, biologically relevant hypothesis or concept to put the results of these experiments into context or warrant their publication. As such, the value of this paper is very limited.

Minor comments:

Figure 1: The images shown are very grainy. The typical elongated morphology of WPBs is difficult to discern and large clusters of granules stacked on top of on another appear to present. With this kind of detail I am not sure how accurate the quantifications are. Is this a PDF conversion issue?

Reviewer #2: The article describes the effect of confluency on WPB numbers in endothelial cells (both HUVEC and ECFCs). The authors compare WPB number in cells seeded at differing densities and cells allowed to reach confluency over a differing period of time.

The effect of confluency on WPB number has been known for a while but there have been contradictory publications which have somewhat muddied the water and this manuscript provides clarity, examines different endothelial types and different ways to get confluent cells and is therefore worthy.

Generally the results are displayed appropriately and the experiments appear sound. Summary data is shown throughout, showing individual experimental points in addition would make the results easier to interpret. More detail in the legends would also help. N numbers are mentioned in the methods but they should also be included in the legends as well as a description of what the error bars represent (SD, SEM etc). Similarly when the WPB numbers are quoted in the text the SD or SEM should also be included to allow readers to assess the veracity of the data.

Fig.1

The legend on the scale bar is really small and should be easier to read. Error bars represent SEM/SD? It would be helpful to have an image at 0h to compare to 100h. It us unclear which timepoint has been shown.

Fig.2

Summary data alone is shown. Individual experimental points would help interpretation (or this data should be shown as supplementary) B and D need error bars ideally.

Fig.3 and 4

Summary data alone is shown, individual data points need to be shown also. The definition of confluency is arbitrary, the authors need to give some idea as to how they delineated confluent vs non confluent with representative images. Microscopy of proteins present at the junctions such as VE Cadherin would make this clearer and allow the readers to gauge the effects and compare the cell types (the authors mentioned differences in cell size etc).

6. PLOS authors have the option to publish the peer review history of their article (what does this mean?). If published, this will include your full peer review and any attached files.

Reviewer #1: No

Reviewer #2: No

---

## [Author Response · Author response to Decision Letter 0]

28 Oct 2022

Dear Editorial Team,

I have added a zipped file of graphpad/csv files, I am wondering if that is sufficient in response to the latest review? Thank you so much!

Alice Liu

---

## [Decision Letter · Decision Letter 1]

9 Nov 2022

Quantitative Analysis of Weibel-Palade Bodies

PONE-D-22-04392R1

Dear Dr. Liu,

We’re pleased to inform you that your manuscript has been judged scientifically suitable for publication and will be formally accepted for publication once it meets all outstanding technical requirements.

Kind regards,

Mária A. Deli, M.D., Ph.D.

Academic Editor

PLOS ONE

Additional Editor Comments (optional):

Reviewers' comments:

Reviewer's Responses to Questions

**Comments to the Author**

1. If the authors have adequately addressed your comments raised in a previous round of review and you feel that this manuscript is now acceptable for publication, you may indicate that here to bypass the “Comments to the Author” section, enter your conflict of interest statement in the “Confidential to Editor” section, and submit your "Accept" recommendation.

Reviewer #1: All comments have been addressed

Reviewer #2: All comments have been addressed

2. Is the manuscript technically sound, and do the data support the conclusions?

Reviewer #1: Yes

Reviewer #2: Yes

3. Has the statistical analysis been performed appropriately and rigorously? 

Reviewer #1: Yes

Reviewer #2: Yes

4. Have the authors made all data underlying the findings in their manuscript fully available?

Reviewer #1: Yes

Reviewer #2: No

5. Is the manuscript presented in an intelligible fashion and written in standard English?

Reviewer #1: Yes

Reviewer #2: Yes

6. Review Comments to the Author

Reviewer #1: The authors have revised some of the text of this manuscript. I agree with their comment that the findings of this paper, which is that the route to confluency does not impact the eventual steady state WPB count, can be of interest to other investigators when planning their experiments.

Reviewer #2: The presentation of the figures are improved and although the authors have not been able to use markers of confluency they have added caveats as to how the data was analysed and the associated implications of this. The primary data is available but is not included as supplementary data and this might be something that will make the findings of the work more impactful.

7. PLOS authors have the option to publish the peer review history of their article (what does this mean?). If published, this will include your full peer review and any attached files.

Reviewer #1: No

Reviewer #2: No

---

## [Editor Report · Acceptance letter]

12 Dec 2022

PONE-D-22-04392R1 

Quantitative Analysis of Weibel-Palade Bodies 

Dear Dr. Liu:

I'm pleased to inform you that your manuscript has been deemed suitable for publication in PLOS ONE. Congratulations! Your manuscript is now with our production department. 

Kind regards, 

on behalf of

Prof. Mária A. Deli 

Academic Editor

PLOS ONE